# Discharge Coefficients of a Heavy Suspension Nozzle

**Carlos Rio-Cano** [1], **Navid M. Tousi** [2], **Josep M. Bergada** [2,*] and **Angel Comas** [3]

[1] Mechanical Engineering Department, Universitat Politècnica de Catalunya, 08034 Barcelona, Spain; carlos.rio-cano@upc.edu

[2] Fluid Mechanics Department, Universitat Politècnica de Catalunya, 08034 Barcelona, Spain; navid.monshi.tousi@upc.edu

[3] Heat Engines Department, Universitat Politècnica de Catalunya, 08034 Barcelona, Spain; angel.comas@upc.edu

[*] Correspondence: josep.m.bergada@upc.edu; Tel.: +34-937398771

**Abstract:** The suspensions used in heavy vehicles often consist of several oil and two gas chambers. In order to perform an analytical study of the mass flow transferred between two gas chambers separated by a nozzle, and when considering the gas as compressible and real, it is usually needed to determine the discharge coefficient of the nozzle. The nozzle configuration analyzed in the present study consists of a T shape, and it is used to separate two nitrogen chambers employed in heavy vehicle suspensions. In the present study, under compressible dynamic real flow conditions and at operating pressures, discharge coefficients were determined based on experimental data. A test rig was constructed for this purpose, and air was used as working fluid. The study clarifies that discharge coefficients for the T shape nozzle studied not only depend on the pressure gradient between chambers but also on the flow direction. Computational Fluid Dynamic (CFD) simulations, using air as working fluid and when flowing in both nozzle directions, were undertaken, as well, and the fluid was considered as compressible and ideal. The CFD results deeply helped in understanding why the dynamic discharge coefficients were dependent on both the pressure ratio and flow direction, clarifying at which nozzle location, and for how long, chocked flow was to be expected. Experimentally-based results were compared with the CFD ones, validating both the experimental procedure and numerical methodologies presented. The information gathered in the present study is aimed to be used to mathematically characterize the dynamic performance of a real suspension.

**Keywords:** discharge coefficients; real compressible flow; Computational Fluid Dynamics (CFD); chocked flow; analytical solutions based on experimental data

## 1. Introduction

Hydro-pneumatic suspensions consist on two or more oil chambers and a couple of gas ones. During the alternative displacement of the suspension, oil and gas flows back and forth between two consecutive chambers often separated by one or several nozzles or valves, therefore generating the smooth suspension displacement characteristic of such devices. Figure 1 introduces a typical heavy vehicle suspension, where several oil and gas chambers can be observed. To mathematically evaluate the dynamic fluid variations associated with the compression and extension of a given suspension, it is required to obtain the dynamic discharge coefficient of the different nozzle shapes separating the gas chambers. The discharge coefficient is defined as $c_d = \frac{\dot{m}}{\dot{m}_t}$, where $\dot{m}$ characterizes the real mass flow flowing through the nozzle, while $\dot{m}_t$ is the mass flow obtained using a theoretical equation. For the suspension configuration presented in Figure 1, the gas chambers consist of a constant volume chamber and a variable volume one. Compressible gas, generally nitrogen, flows between two chambers through a narrow passage of constant cross-section, a nozzle. In the suspension of the present study, the pressure in both nitrogen chambers is time-dependent, which is the reason why it is important to determine the discharge coefficient variation for a real compressible flow under transient conditions.

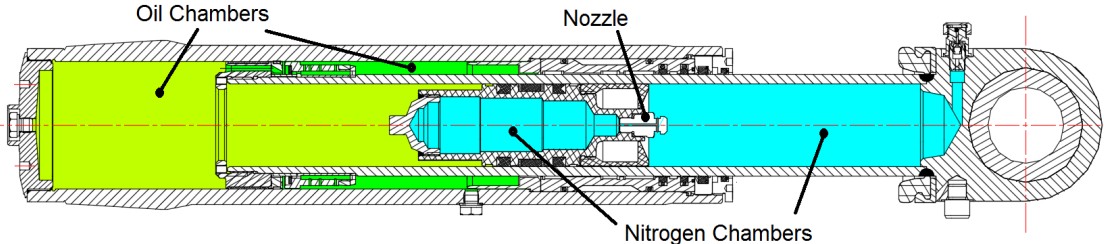

**Figure 1.** Scheme of a typical heavy vehicle suspension.

The main advantage of knowing the discharge coefficient of a given nozzle is that it allows to determine the real mass flow through the nozzle via employing theoretical equations. From the existing literature, several traditional experimental procedures [1,2] are described to experimentally determine the discharge coefficients on nozzles under real flow compressible conditions. The accuracy of these methods is good for a limited range of pressures, but it is jeopardized in applications involving high pressure metering. One of the most relevant early papers on non-ideal gas flow through orifices is the work undertaken by Johnson [3], where an expression for one-dimensional real flow through nozzles and based on the Beattie-Bridgeman equation was developed. The specific heat ratio was considered to be constant. Air at room temperature and for 10 MPa pressure differential was used as working fluid, and he observed a deviation of 3.5 percent for critical flow in nozzles when employing real versus ideal gas equations. Bober and Chow [4], using Methane as working fluid and for a pressure differential of 23 MPa between chambers, compared the ideal and real gas flow performance through a venturi-shaped nozzle using the Redlich-Kwong equation. Under choked flow conditions, the difference between ideal and real gas models was about 20%. Based on these early papers, it became clear that real gas effects had to be considered if precision metering was required.

Kouremenos et al. [5] and Kouremenos and Antonopoulos [6], based on the Lee-Kesler and Redlich-Kwong equations of state, described a constant entropy process via using three isentropic exponents. A set of simulations and experimental measurements of real compressible flow though convergent divergent nozzles, at very high pressure differentials, were recently undertaken by Kim et al. [7,8] and Nagao et al. [9,10]. They observed that, for a given range of Reynolds numbers, the discharge coefficient exceeded unity. This fact was previously reported by Nakao [11] experimentally. They realized that the molecules' vibrational energy had to be considered in the non-equilibrium thermodynamic process. Working with hydrogen, Ding et al. [12] observed the discharge coefficient was not just dependent on the Reynolds number but also on the throat diameter, stagnation pressure and stagnation temperature. They realized that the compressibility factor (Z) was changing in opposite direction than the discharge coefficient and concluded that the compressibility factor was likely the most important parameter when studying the discharge coefficient. They also noticed that, due to real gas effects, the fluid density at the nozzle throat became smaller than the theoretical one. In most of the research undertaken previously, the theoretical work was not supported by an experimental method, which could allow working directly with the experimental data. These aspects were covered in Reference [13], where, for constant section short nozzles, an expression defining the discharge coefficient was developed using an experimental method and a gas flow model based on the Lee-Kesler equation of state [14]. Experimental results were compared with the ones obtained from the new developed equation, observing that, for tests performed using nitrogen up to 7.6 MPa pressure differential, a good correlation was obtained.

Computational Fluid Dynamic (CFD) applications are gaining reliability every day. The consideration of the gas as real and compressible, under sonic or supersonic flow conditions, is still not fully extended. In reality, turbulent models quite often have some difficulties in dealing with such kind of flows. Nevertheless, there are many industrial applications where sonic, and even supersonic, flow is used. CFD simulations on compressible flow through valves, whether purging or relief ones, have recently been performed

in Reference [15,16]. The modeling of flow ejectors under sonic flow conditions was considered in Reference [17–21]. Some of the recent papers simulating compressible flow conditions at high Mach numbers inside nozzles are Reference [22–25].

From all these studies, it is particularly relevant to highlight the work done by Farzaneh-Gord et al. [15], where they numerically evaluated the exit flow of natural gas through a purging valve, during its opening time. They considered the gas as real and compressible, being the maximum pressure differential of 3.5 MPa. As a turbulent model, the standard $k - \epsilon$ was selected. They concluded supersonic flow was to be expected at the pipe outlet. García-Todolí et al. [16] performed CFD simulations on air valves under compressible flow condition. They showed how CFD models are efficient to represent the behavior of air entering and leaving the valve. The maximum pressure differential studied was 0.1 MPa. In their analysis, they used the realizable $k - \epsilon$ turbulent model and their results matched very well with the experimental data. Mazzelli et al. [17] performed numerical and experimental analyses in order to check the effectiveness of the commonly-used computational techniques when predicting ejector flow characteristics at supersonic flow conditions. For the numerical part, they considered the working fluid as an ideal gas. They tested different Reynolds-averaged Navier–Stokes (RANS) turbulent models, among them, the $k - \omega$ SST and $k - \epsilon$ realizable ones, and observed that, in general, all turbulent models generated very similar results, although epsilon-based models were more accurate at low pressure differentials (around 0.2 MPa). The different pressure differentials they evaluated were of 0.2, 0.35, and 0.5 MPa. On the other hand, they stated that the main differences between the numerical and experimental results appeared when comparing 2D and 3D models. Lakzian et al. [18] performed a compressible 2D RANS simulation on an air ejector pump. In their analysis, they assumed the working fluid as ideal gas and the walls were treated as adiabatic. Pressure differentials of 0.5, 0.6, 0.7, and 0.8 MPa were considered. They used a $k - \epsilon$ realizable turbulent model along with a wall function and a very good agreement with experimental data was obtained. They concluded that the main sources of entropy are the mixing and normal shock occurred in the mixing chamber and diffuser, respectively. Arias and Shedd [21] used CFD to develop a 3D model of compressible flow across a venturi in which obstacles were located inside. Air was considered as compressible and was treated as ideal gas. The turbulent model they used was RNG $k - \epsilon$ and the maximum pressure differential was about 0.1 MPa. The results showed that the obstacles located at the converging nozzle of the venturi causes negligible pressure losses, while other obstacles that generate wakes in the flow are responsible for the largest pressure drop. Discharge coefficients of critical nozzles used for flow measurement under compressible flow conditions were evaluated by Ding et al. [22]. Fluid was considered as real, the standard $k - \epsilon$ turbulent model with a wall function was employed in all simulations. Nozzle roughness was considered, being the maximum pressure differential between nozzle inlet-outlet of 120 MPa. They observed that, when the nozzle roughness was very small, and for pressure differentials until 1 MPa, the effect on the discharge coefficient was negligible. Sonic and supersonic flow inside micro/nanoscale nozzles was studied by Darbandi and Roohi [23]. They used a density-based solver (rhoCentralFoam) employed in OpenFOAM. Second order spatial discretization scheme along with a first order Euler-scheme for time integration were implemented. They observed supersonic flow was impossible to set in nanoscales once Knudsen number exceeded a given value. Zhao et al. [24] numerically studied the fuel flow in a nozzle considering the fuel compressibility. They used the RANS method with a Realizable $k - \epsilon$ turbulent model and they investigated the effect of injection pressure on the fuel flow under fuel compressibility conditions. They concluded that the nozzle discharge coefficient for compressible flow was larger than when fluid was considered as incompressible.

According to the authors knowledge, the nozzle configuration studied in the present paper, which has a T shape, has not been previously studied under real gas compressible flow conditions, and just the work done by Farzaneh-Gord et al. [15] presents some similarities. In fact, under incompressible flow conditions, a similar shape was studied by

Reference [26,27], where it was stated the discharge coefficient was highly dependent on the flow direction. The present study consists of the following parts: initially, the test rig employed to do all experimental tests is introduced, and then the mathematical equations used to analytically determine the flow parameters are presented. In a third stage, the Computational Fluid Dynamics (CFD) methodology employed to numerically evaluate the compressible flow between the two tanks is introduced. Next, the experimentally-based and numerical results are presented and compared. Finally, the discharge coefficients as a function of the Reynolds number, and for both flow directions, are presented and discussed, and the paper ends with the conclusions.

## 2. Experimental Test Rig

Since the primary idea in the present paper was to experimentally determine the directional dynamic discharge coefficients for a real gas, air, the test rig introduced in Figure 2 was created. Figure 2a,b, respectively, show a general view of the test rig and the two reservoirs. Figure 2c presents a schematic view of the two reservoirs central section with the different transducers employed. Apart of the two reservoirs, the test rig consisted of a stopper cylinder, which was employed to displace the shutter valve located inside the large reservoir; see Figure 2c. When this valve was closed, it prevented the fluid from flowing between the two reservoirs, allowing to pressurize each of them independently. The volume of each reservoir was of 2288.48 and 700.18 cm$^3$. Notice that the volume of the large reservoir was slightly increasing as the shutter valve was opening. Therefore, in order to know at each instant which was the real volume, a position transducer was attached to the stopper cylinder. The pressure multiplier in conjunction with the pressurize-depressurize valve were connected to the large reservoir and allowed to set the required pressure in the reservoir chambers. A static and a dynamic pressure transducer were connected to each reservoir; this was done due to the uncertainty of the static transducer in properly measuring the dynamic pressure variations. The fluid temperature was aimed to be measured by several dynamic thermocouples placed on the internal reservoirs walls; one of them was located in the large reservoir, and three were placed on the smaller one, and then larger temperature variations were expected in there.

It is important to notice that the dynamic thermocouples were welded to the internal walls of the reservoirs; therefore, the measured temperature was, in reality, the internal wall temperature, which may not be exactly the same as the fluid temperature, specially under dynamic conditions. Considering known the dynamic temperature at the upstream reservoir, Kagawa et al. [2] identified the susceptibility of the pressure response to temperature changes in a reservoir at natural conditions, and suggested the use of an upstream isothermal chamber to guarantee no temperature variation. This idea appears to be a good solution since it reduces the number of variables, and assuming pressure is a known variable, it would be possible to mathematically determine the temperature, heat transfer, and the mass flow downstream by means of integration. The main difficulty lies in achieving no upstream temperature variation during the discharge, Kagawa et al. [2] suggested stuffing extremely thin steel wool or copper wire in the upstream reservoir to obtain isothermal conditions. For the present experimental test rig, and in order to tend to achieve isothermal conditions during experimentation, both reservoirs walls were constructed with a thickness of 35 mm.

The main characteristics of the different transducers were: the static pressure sensors were from Keller series model 21/21PRO, capable of measuring pressures of 10 MPa and having a resolution of 100 Pa. To properly evaluate the dynamic pressure, Kistler transducers model 601A were used; their resolution was of 100 Pa, and the time response was of 1 μs. Dynamic temperature was measured using low inertia Nammac thermocouples model E6-20, and their resolution was of 0.01 °C. The transducer used to measure the position of the shutter valve was a LVDT type, model CGA-2000 from TE connectivity, and its resolution was of 0.001 mm. The dynamic variables were recorded, thanks to an in-house LabVIEW program specifically developed for this application. Table 1 introduces

the initial absolute pressure on both reservoirs employed in each experimental test, as well as in each CFD case. The main dimensions of the T shape nozzle are defined in Figure 2c. The constant section nozzle diameter of both the horizontal and vertical nozzle branches was $d = 1.5$ mm, the horizontal branch length ($L_1$) was $L_1 = 29.5$ mm, and the length of the vertical branch ($L_2$) was $L_2 = 10$ mm.

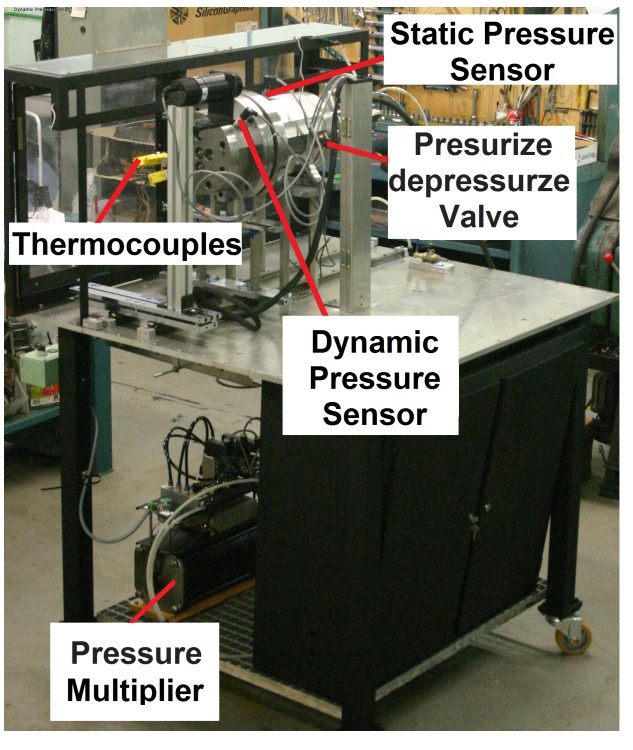

(**a**)

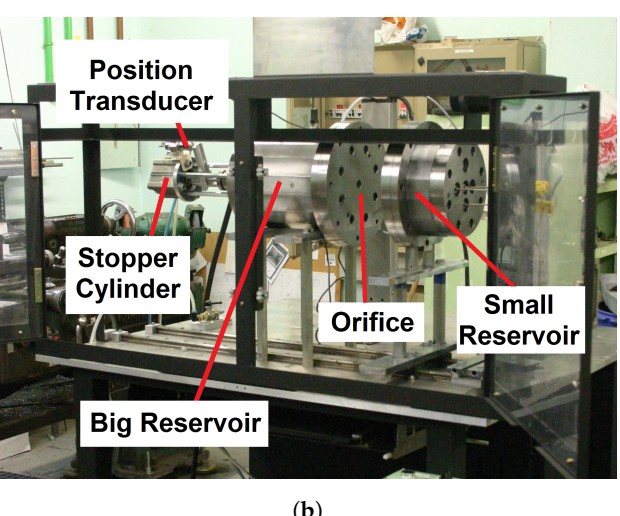

(**b**)

(**c**)

**Figure 2.** (**a**) Test rig main view. (**b**) The two reservoirs with their pressure and temperature transducers; (**c**) schematic view of the reservoirs and the transducers localization.

The process followed to perform the simulations was a function of the flow direction, and it was depending on which of the two reservoirs was initially pressurized. In other words, it depends on if the discharge was from the large to the small reservoir (L-to-S) or from the small to the large (S-to-L) one. It is important to notice that the T shape nozzle was always kept in the same position, regardless of the flow direction.

**Table 1.** Initial absolute pressure at both reservoirs for all different experimental tests performed and for all Computational Fluid Dynamic (CFD) simulations undertaken.

| Pressure large reservoir (MPa) | 0.4 | 0.6 | 0.9 | 1.1 | 1.2 | 0.1 | 0.1 | 0.1 | 0.1 | 0.4 | 0.4 CFD | 1.1 CFD | 0.1 CFD | 0.4 CFD |
|---|---|---|---|---|---|---|---|---|---|---|---|---|---|---|
| Pressure small reservoir (MPa) | 0.1 | 0.1 | 0.1 | 0.1 | 0.4 | 0.4 | 0.6 | 0.9 | 1.1 | 1.2 | 0.1 CFD | 0.1 CFD | 0.6 CFD | 1.2 CFD |

The measurements done when the large reservoir was pressurized started with the shutter valve open and both reservoirs at the atmospheric pressure, 0.1 MPa absolute pressure. Under these conditions, the shutter valve was closed, and the large reservoir was filled with air, passing through the pressurize-depressurize valve, until reaching the required pressure. Then, the pressurize-depressurize valve was closed. The last step consisted of opening the shutter valve, which was accomplished by pressurizing the stopper cylinder, and the flow was allowed to go from the large to the small reservoir. The stopper cylinder position, the static, and dynamic pressure, as well as the internal walls temperature, on both reservoirs, were recorded using an external computer, and thanks to a labVIEW program. For tests at which the fluid was going from the small reservoir to the large one, the same procedure was used but initially both reservoirs were pressurized at the pressure required for the small reservoir. After closing the shutter valve and using the pressurize-depressurize valve, the pressure at the large reservoir was decreased until obtaining the one needed. It is important to highlight at this point that each test was done ten times, and the resulting curves presented in the results section are the average value of the ten measurements done for each variable.

### 3. Mathematical Equations and Analytical Process Followed to Determine the Physical Variables

From the experimental tests, it was soon realized that, as the temperature transducers were welded to the reservoirs internal walls, in reality, they were measuring the temperature of the wall and not the fluid temperature at each reservoir. The measured temperature was almost constant in both reservoirs, and it was used to instantaneously estimate the heat transferred through the walls. Therefore, providing that the only trustworthy dynamic information in both reservoirs was the temporal pressure evolution, the methodology employed to determine the mass flow between reservoirs was based on the following equations, developed by the authors in a former paper [13]. Equation (1) was obtained from the application of the energy equation in the upstream reservoir, it characterizes the temporal mass variation in the upstream reservoir, $\frac{dm_u}{dt}$, as a function of the tank's temperature $T_u$, the heat transferred to the fluid $\frac{dQ_u}{dt}$, the upstream pressure temporal variation $\frac{dp_u}{dt}$, the variation of the compressibility factor versus the temperature and specific volume $\frac{\partial Z_u}{\partial T}$; $\frac{\partial Z_u}{\partial v}$, the mass of the fluid in the upstream reservoir $m_u$, the enthalpy $h_u$, and the internal energy $u_u$ associated with the upstream fluid. The heat transferred across the walls of the reservoir was estimated based on the Fourier equation $Q = -\lambda \frac{dT}{dx}\big|_{x=0}$, being the value of the thermal conductivity $\lambda = 54\left(\frac{W}{mK}\right)$. Equations (2) and (3) arise from the differentiation of the real gas equations applied to the upstream and downstream reservoirs, $p_u \forall_u = Z_u m_u R_u T_u$; $p_d \forall_d = Z_d m_d R_d T_d$, and they link the pressure, volume,

temperature, mass flow, and compressibility factor temporal variations existing in the respective upstream and downstream reservoirs. Equation (4) simply characterizes the mass transfer balance. In all these equations, sub-indices $u$ and $d$ stand for upstream and downstream, respectively.

$$\frac{dm_u}{dt} = \frac{\frac{1}{p_u}\frac{dp_u}{dt} - \frac{1}{m_u c_v}\left[\frac{1}{T_u} + \frac{1}{Z_u}\left(\frac{\partial Z_u}{\partial T}\right)_v\right]\frac{dQ_u}{dt}}{\frac{1}{m_u} - \frac{1}{Z_u}\left(\frac{\partial Z_u}{\partial v}\right)_T \frac{\forall_u}{(m_u)^2} + \frac{1}{m_u c_v}\left[\frac{1}{T_u} + \frac{1}{Z_u}\left(\frac{\partial Z_u}{\partial T}\right)_v\right]\left[RT_u^2\left(\frac{\partial Z_u}{\partial T}\right)_v - (h_u + u_u)\right]}, \tag{1}$$

$$\frac{dT_u}{dt} = \frac{T_u}{p_u}\frac{dp_u}{dt} + \frac{T_u}{\forall_u}\frac{d\forall_u}{dt} - \frac{T_u}{m_u}\frac{dm_u}{dt} - \frac{T_u}{Z_u}\frac{dZ_u}{dt}, \tag{2}$$

$$\frac{dT_d}{dt} = \frac{T_d}{p_d}\frac{dp_d}{dt} + \frac{T_d}{\forall_d}\frac{d\forall_d}{dt} - \frac{T_d}{m_d}\frac{dm_d}{dt} - \frac{T_d}{Z_d}\frac{dZ_d}{dt}, \tag{3}$$

$$\frac{dm}{dt} = \frac{dm_d}{dt} = -\frac{dm_u}{dt}. \tag{4}$$

An in-house computer program was created to solve the preceding Equations (1)–(4) with a Runge-Kutta method based on DVERK from the International Mathematics and statistics library, (IMSL). The air density was determined every time step using the Lee-Kesler equation iteratively as performed by Plocker and Knapp [28]. According to this equation, the compressibility factor as a function of the reduced parameters, can be expressed as:

$$Z^{(r)} = \left(\frac{P_r v_r}{T_r}\right) = 1 + \frac{B}{v_r} + \frac{C}{v_r^2} + \frac{D}{v_r^5} + \frac{c_4}{T_r^3 v_r^2}\left(\beta + \frac{\gamma_1}{v_r^2}\right)e^{\frac{-\gamma_1}{v_r^2}}. \tag{5}$$

The parameters B, C, D, $c_4$, $\beta$, and $\gamma_1$, for various gases, can be determined from Reference [5]. $P_r$, $T_r$, and $v_r$ stand for reduced pressure, reduced temperature, and reduced specific volume, respectively.

According to Lee and Kesler [14], the compressibility factor can be defined as:

$$Z = Z^{(0)} + \frac{\omega}{\omega^{(R)}}\left(Z^{(R)} - Z^{(0)}\right), \tag{6}$$

where $Z^{(R)}$ and $Z^{(0)}$ are the compressibility factors for a reference fluid and simple fluid, respectively, while $\omega^{(R)}$ and $\omega$ stand for the acentric factors of the reference and working fluids.

Considering known the pressure and temperature in a given location and time, the following steps were used to calculate the fluid compressibility factor. Initially, the reduced upstream pressure and temperature ($P_r$ and $T_r$) were obtained based on the working fluid critical properties ($P_c$; $T_c$) and the values of the pressure and temperature. When introducing the values of $P_r$ and $T_r$ in Equation (5), introducing, as well, the values of the parameters given for a simple fluid and obtained from reference [5], the value of $v_r$ could be determined. Substituting the value of $v_r$ in the same Equation (5), the corresponding compressibility factor for a simple fluid $Z^{(0)}$ was obtained. Following the same procedure just described, but using the values of the parameters characterizing the reference fluid, which, for the present study, was n-octane, and values obtained from Reference [5], the value of $Z^{(R)}$ was determined. Substituting the compressibility factors $Z^{(0)}$ and $Z^{(R)}$ in Equation (6) and considering the acentric factor values $\omega^{(R)}$ = 0.3978 and $\omega$ = 0.039, the compressibility factor for the working fluid could finally be obtained. This procedure allows to determine the compressibility factor and the fluid density at any position and time, and just the values of the pressure and temperature are required at the generic location where the information is needed.

To be able to determine the instantaneous mass at each reservoir, the pressure evolution was measured in both reservoirs at any time; the volume of both reservoirs was also

known, and the fluid temperature, as well as the compressibility factor, were estimated based on the previous equations. The variation of the fluid mass between two consecutive time steps allowed to calculate the instantaneous mass flow leaving one reservoir and entering the other one. The only problem associated with this methodology was that the fluid temperature had to be estimated. As previously defined by Kagawa et al. [2] and Comas et al. [13], if the reservoirs were large enough, the fluid temperature was likely to remain constant. Yet, which were the required dimensions to fulfill this condition, for each particular case, was not clearly stated.

Based on the previous information, the instantaneous space averaged fluid velocity at the nozzle minimum section $S = \frac{\pi d^2}{4}$ was determined as presented in Equation (7). The fluid velocity at the critical section was determined based on the experimentally-based mass flow $\dot{m}$, the nozzle section $S$, and the fluid downstream density $\rho_d$.

$$\vartheta = \frac{\dot{m}}{S\rho_d}. \tag{7}$$

To determine the Mach number, the sound speed was initially obtained from Equation (8), when substituting Equations (7) and (8) in Equation (9), the Mach number at the nozzle critical section was obtained.

$$c^2 = \left(\frac{\partial p}{\partial \rho}\right)_s = -v^2 \left(\frac{\partial p}{\partial v}\right)_s, \tag{8}$$

$$M = \frac{\vartheta}{c}. \tag{9}$$

On the other hand, and due to the fact that the pressure differential between both reservoirs was relatively small, as in Table 1, the following equation was employed to calculate the theoretical mass flow.

$$\dot{m}_t = \frac{\pi D^2}{4} \sqrt{\frac{2\gamma}{\gamma - 1} p_u \rho_u \left( \left(\frac{p_d}{p_u}\right)^{\frac{2}{\gamma}} - \left(\frac{p_d}{p_u}\right)^{\frac{\gamma+1}{\gamma}} \right)}. \tag{10}$$

The instantaneous discharge coefficient was determined at each time step by comparing the real and theoretical mass flows. Actually, the discharge coefficient at each time step was obtained according to Equation (11).

$$c_d = \frac{\dot{m}}{\dot{m}_t}, \tag{11}$$

where $\dot{m}$ is the real mass flow obtained based on the temporal variation of the mass in the upstream reservoir, which was determined from the experimental upstream and downstream pressure evolution, the initial fluid temperature, and after calculating the compressibility factor, as well as the fluid temperature evolution, at each time step. $\dot{m}_t$ is the mass flow obtained via using Equation (10).

At each instant, the Reynolds number was determined using the following equation.

$$Re = \frac{4\dot{m}}{\pi D \mu}, \tag{12}$$

where $\mu$ is the fluid dynamic viscosity.

## 4. Dynamic Computational Fluid Dynamic Simulations

In order to be able to analyze the dynamic flow evolution between the two reservoirs, several 3D Computational Fluid Dynamic (CFD) simulations were undertaken. The working fluid was air, and it was considered as ideal and compressible. Some recent papers in which the fluid was considered as ideal and compressible and working under similar

pressure differentials are Reference [17,18]. In the present paper, the simulations were performed under dynamic conditions, therefore matching the experimental test conditions.

Figure 3 shows the two reservoirs separated by the T shape nozzle, and the dimensions of both reservoirs and the T shape nozzle were the same as the ones used in the experimental tests. The only difference was the shutter valve needed in the experimental test rig, as in Figure 2c, which was not required in the CFD simulations. The mesh employed was generated using GMSH, and it was unstructured and consisted of 126,633 cells. The OpenFOAM software was used for all 3D simulations, and finite volumes is the approach OpenFOAM uses to discretize Navier–Stokes equations. The solver rhoCentralFoam was used for all tests, and the spatial discretization was set to second order being the first order Euler scheme the one used for time discretization. The maximum Courant number was kept below 0.8, being the time step around $5 \times 10^{-8}$ s. Turbulence intensity was set to 0.05% in all cases. The realizable $k - \epsilon$ turbulent model, along with a wall function, as previously used by Lakzian et al. [18], were employed in all the simulations. The maximum $y^+$ on the wall of the nozzle was about 90. Volumetric Dirichlet pressure and temperature boundary conditions were initially set in both reservoirs, Newman boundary conditions for pressure and temperature were set in all walls, and Dirichlet boundary conditions for velocity were established in all walls. Regarding the heat transfer, all walls were set as adiabatic. To compare with the experimental results, four simulations were performed; in two of them, the flow was going from the large to the small reservoir, and the respective L-to-S reservoirs pressures were 0.4–0.1 MPa and 1.1–0.1 MPa. In the other two simulations, the fluid was flowing from the small to the large reservoir, being the S-to-L reservoirs pressures, respectively, of 0.6–0.1 MPa and 1.2–0.4 MPa; see Table 1.

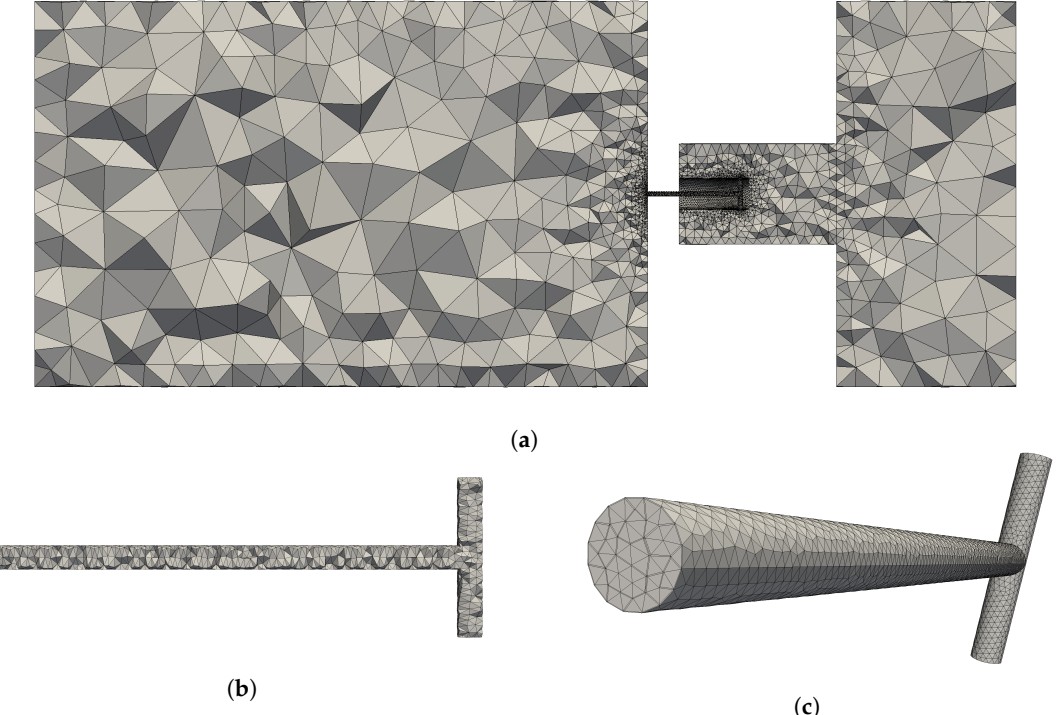

(a)

(b)

(c)

**Figure 3.** (**a**) Reservoirs main mesh; (**b**) T shape nozzle section mesh; (**c**) T shape nozzle mesh general view.

## 5. Results and Discussion

In the present section, initially the measured temporal pressure evolution inside the reservoirs is compared with the ones obtained from the CFD simulations. The same is later being done with the temporal temperature at the upstream reservoirs. Next, the time-dependent mass at each reservoir is also compared between CFD and experimentally-based results, which is followed by the temporal mass flow comparison. The time-dependent

Mach numbers at the respective critical sections and the discharge coefficients versus the Reynolds number are presented next. At the end of this section, a figure showing the flow inside the nozzle and for both flow directions is introduced. In this figure, the critical sections where the flow becomes sonic and the locations where supersonic flow is to be expected are clearly stated.

Figure 4 presents the temporal pressure variation measured in both reservoirs for the different initial pressure differentials introduced in Table 1. Each curve is, in reality, the average one obtained after performing each test ten times. Although not presented in Figure 4, the standard deviation of each point was smaller than 1% for all tests performed. From Figure 4a, it is observed the discharge lasts about two seconds, regardless of the initial pressure differential existing between the two reservoirs. In fact, the time needed to complete the discharge suffers an increase of about 22.5% when comparing the discharge from 0.4 to 0.1 MPa with the 1.1 to 0.1 MPa one. This phenomenon is clearly understandable then the higher the upstream pressure the higher is the mass to be transferred from one reservoir to the other. Notice that the initial mass of fluid in the downstream reservoir is the same for all cases presented in Figure 4a, except the case at which the initial downstream pressure is of 0.4 MPa, being the upstream pressure of 1.2MPa. This case shows clear differences versus the rest of the discharges; then, the time required to complete the discharge is about 10% shorter than the one needed to complete the discharge when the upstream/downstream reservoir pressures were 0.4 and 0.1 MPa, respectively. In fact, the discharge time is directly related to the initial fluid density ratio $\rho_{upstream}/\rho_{downstream}$ between reservoirs, given the rest of the parameters, reservoirs volumes, and initial fluid temperature as constant, the smaller the initial upstream/downstream density ratio $\rho_{upstream}/\rho_{downstream}$, and the shorter the discharge time.

From the observation of the temporal pressure decay when the flow goes from the small reservoir to the large one, as in Figure 4b, it is realized that the discharge time obeys to the same upstream/downstream density rule just presented. It is interesting, as well, to observe that, when comparing discharge times for the same pressure differential and opposite flow directions, the discharge time is larger when the fluid goes from the small to the large reservoir. This is likely linked to the resistance the T shape nozzle is presenting when the fluid flows in such direction. The time the flow remains under chocked conditions it is expected to depend on such resistance. This point is to be clarified in the remaining part of the paper. Figure 4 also compares the pressure decay and increase measured experimentally with the ones obtained from the CFD simulations, two cases are compared for each flow direction. The comparison shows a very good agreement, generating the same discharge times and final pressures as the ones measured experimentally. Small discrepancies are observed in the final pressure values when the flow goes from the small to the large reservoir, a discrepancy of 4.4% is observed for a discharge from 1.2 to 0.4 MPa, and the variation reaches 11.3% for a discharge from 0.6 to 0.1 MPa. Such relatively small discrepancies are understandable when considering that, in the CFD simulations, the process is considered as adiabatic, the fluid is considered as ideal, and the large volume remains constant.

Figure 5 presents the fluid temperature evolution in both reservoirs and for the four cases numerically evaluated. When the discharge is from the large to the small reservoir, as in Figure 5a, the fluid temperature on the large reservoir suffers a decrease of less than 20°, and the increase of the fluid temperature in the small reservoir lies between 55° and 85°; such a large increase is perfectly understandable when considering the reduced volume of this particular reservoir and that the walls are considered adiabatic. Notice, as well, that the temperature decrease and increase are directly dependent on the pressure ratio between reservoirs. When the flow goes from the small to the large reservoir, as in Figure 5b, the temperature decrease in the small tank oscillates between 50° and 70°, and a maximum temperature rise of around 30° is observed in the large reservoir.

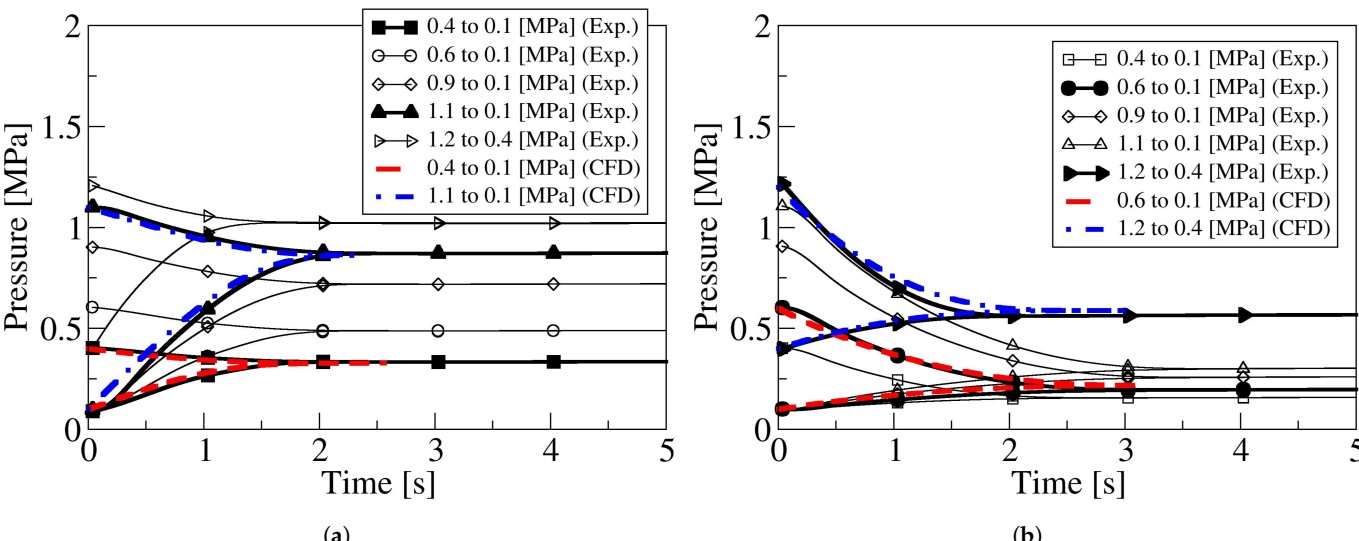

**Figure 4.** Measured temporal pressure variation in both reservoirs, comparison between experimental and CFD results. (**a**) Flow from large to small reservoir. (**b**) Flow from small to large reservoir. Five different pressure decays are considered for each flow direction; see Table 1.

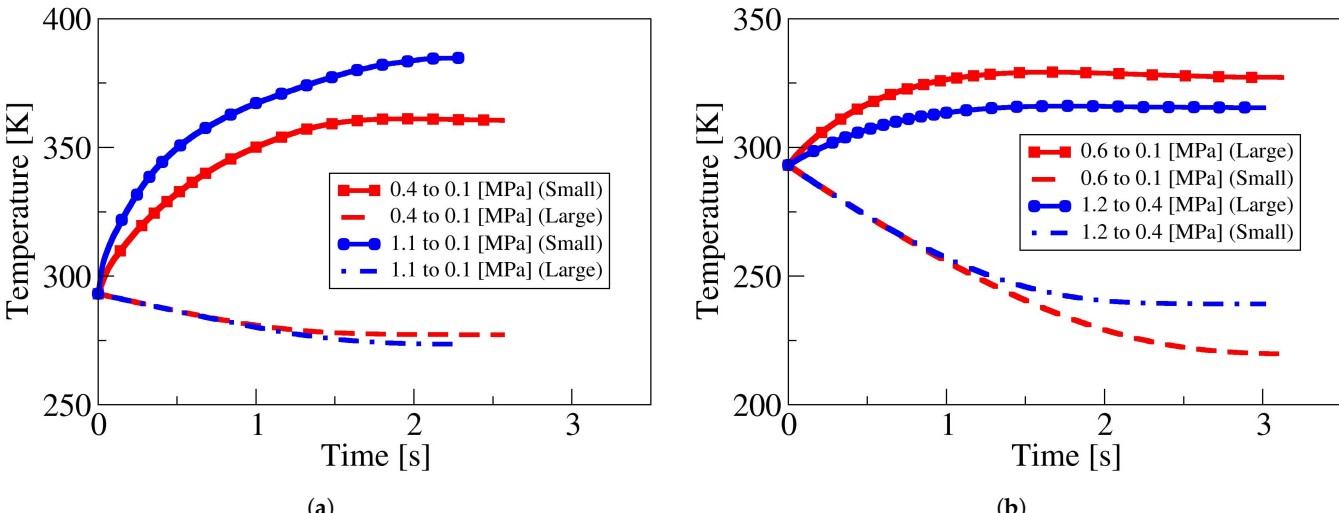

**Figure 5.** Numerical temperature evolution in the upstream and downstream reservoirs. (**a**) The flow goes from the large to the small reservoir. (**b**) The flow goes from the small to the large reservoir.

A point which needs to be considered, and which could help to explain why the discharge from the small to the large reservoir takes longer than the one in the opposite direction, is that, regardless of the flow direction, the temperature variation at the small reservoir is several times larger than the one observed at the large one. Another possible explanation needs to be found in the possible existence of a flow restriction under these conditions, therefore reducing the effective flow section. In fact, the most plausible explanation is likely to be the different nozzle resistance the fluid is facing when flowing in opposite directions. These hypotheses will be analyzed in the remaining part of the paper.

Figure 6 introduces the experimentally-based fluid temperature temporal variation on both reservoirs for the two flow directions and for all pressure differentials evaluated; see Table 1. Figure 6a,b characterize the temperature decrease in the large and small reservoirs when the flow goes from the large to the small and from the small to the large reservoirs, respectively. The fluid temperature evolution in both reservoirs obtained from the CFD simulations is also presented for comparison.

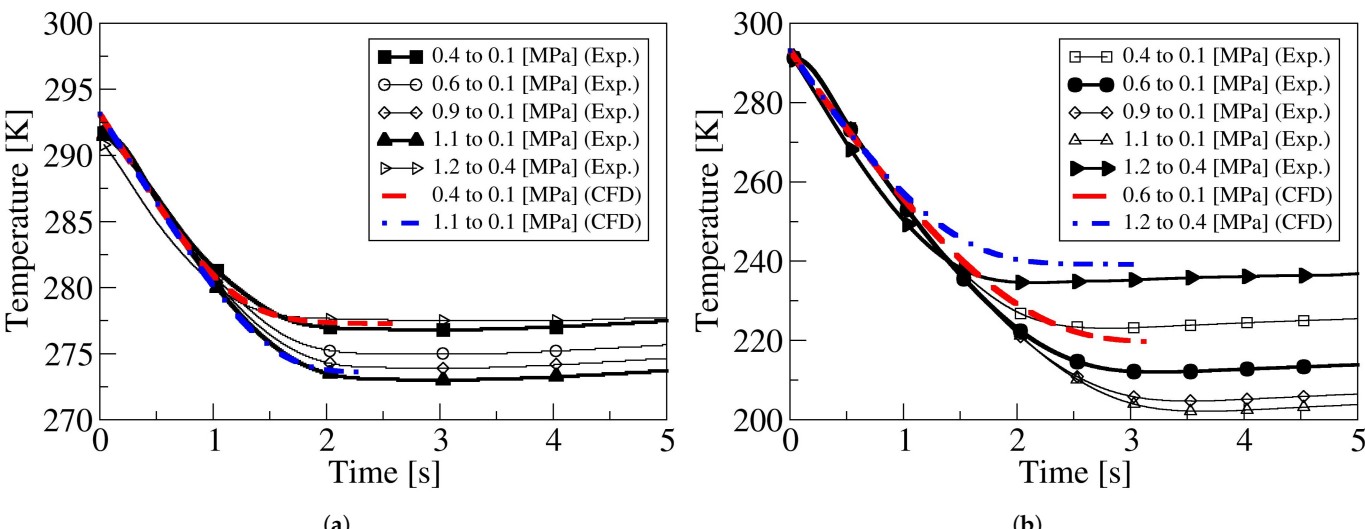

**Figure 6.** Experimentally-based and numerical temperature decay in the upstream reservoir. (**a**) The flow goes from the large to the small reservoir. (**b**) The flow goes from the small to the large reservoir.

The first thing to be observed is that the fluid temperature decay is proportional to the initial pressure ratio between reservoirs, and the higher the pressure ratio, the higher the fluid temperature decay in the upstream reservoir. As previously observed, the temperature drop is particularly high in the small reservoir. Temperature decreases of over 50° are observed in the small reservoir, and such decrease is of less than 20° in the large one. When comparing the temperature evolution experimentally-based with the CFD one, a particularly good agreement is observed in the large reservoir, and a maximum difference between experimental and numerical results of about 3.3% is observed in the small reservoir when the discharge is from 0.6 to 0.1 MPa. The experimentally-based results generate final temperatures slightly lower than the ones obtained via CFD simulations. As the walls were assumed adiabatic in the CFD simulations, Figure 6 confirms this assumption; then, the heat transferred through the walls appears to be negligible.

Based on the experimental pressure temporal evolution and the calculated temperature, the temporal mass variation at each reservoir for both flow directions and for all different pressures studied is presented in Figure 7. The same figure presents, as well, the mass decay/increase obtained via CFD. Figure 7a,b introduce the reservoirs temporal mass variation when the air flows from the L-to-S and S-to-L reservoirs, respectively. As previously observed, when the discharge is from 1.2 to 0.4 MPa, the discharge time is minimum, and this is due to the small density ratio associated to the fluid. Regardless of the flow direction, the curves representing the temporal mass variation on both reservoirs are, for the discharge initial first second, having a constant pendent, but, during the next 1.5 discharge seconds, the curves are rounded. The constant pendent is likely to indicate chocked flow conditions. The curves of the Mach number versus time should clarify this hypothesis.

The instantaneous mass flow flowing between the two reservoirs for all pressures studied and for both flow directions, is presented in Figure 8. Notice that the information presented in this figure was directly extracted from Figure 7. For each pressure ratio, Figure 7 presents two curves, representing the mass decrease in one reservoir and the mass increase in the other; therefore, each of the mass flow curves could be obtained twice, considering the mass decrease and increase in the respective reservoirs. Since both mass flow curves were almost identical, in Figure 8, just the curves representing the mass flow decrease in the upstream reservoir are presented. Figure 8a characterizes the mass flow between reservoirs when the fluid is going from the large to the small reservoir. Notice that, as the pressure ratio increases, the mass flow also increases. In reality, this mass flow increase associated with the pressure ratio increase is due to the upstream

fluid density increase. It is as interesting to see that, as the pressure ratio increases, the overall discharge time, and the time at which the flow remains under sonic conditions, also increases; Figure 9 shall further clarify this point. Figure 8b presents the mass flow for the fluid going from the small to the large reservoir. It is interesting to realize that, regardless of the pressure ratio evaluated, the discharge time lasts almost a second longer than when the flow goes in the opposite direction. As already observed in Figure 8a, for a discharge from 1.2 to 0.4 MPa, the pendent of the mass flow curve is much higher than for the rest of the cases evaluated, clearly showing that the initial downstream density plays an important role when considering the discharge temporal evolution and final time, and such time decreases with the density ratio decrease.

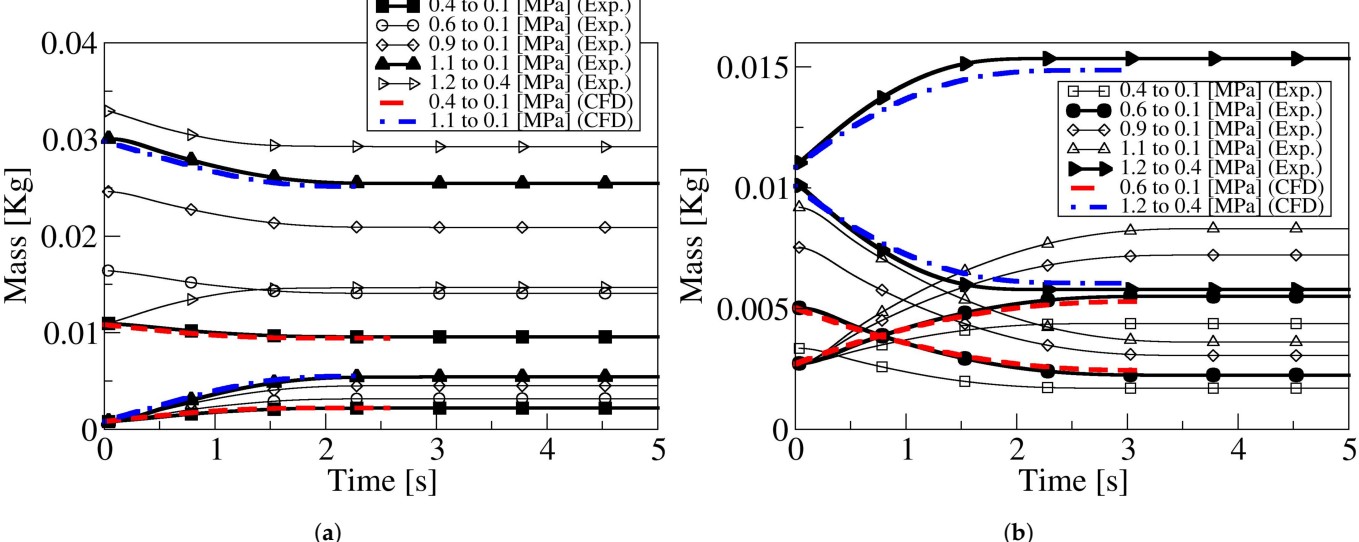

(a)

(b)

**Figure 7.** Temporal mass variation in each reservoir based on experimental data and for all pressures studied. (**a**) Flow from large to small reservoir. (**b**) Flow from small to large reservoir.

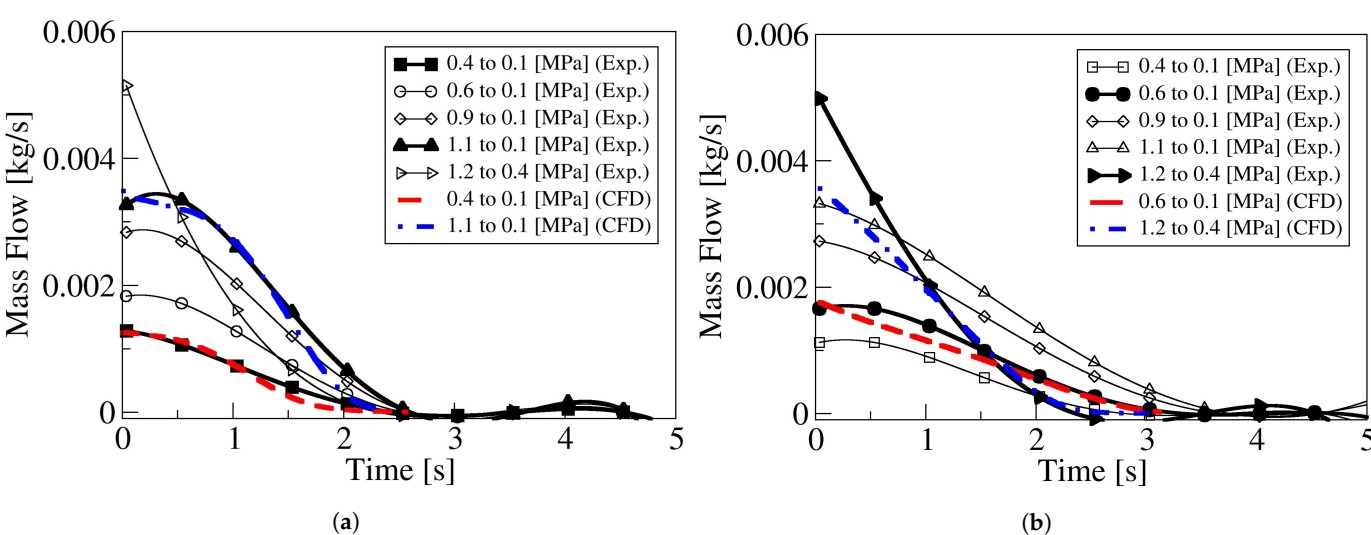

(a)

(b)

**Figure 8.** Mass flow between reservoirs, based on experimental data and for all pressures studied. (**a**) Flow from large to small reservoir. (**b**) Flow from small to large reservoir.

Figure 8 also compares the mass flow obtained experimentally with the numerical one, and the agreement appears to be very good for all cases studied; just when the discharge is from 1.2 to 0.4 MPa, and the flow goes from the small to the large reservoir, the pendent of the mass flow during the initial 0.5 s shows some discrepancy. In fact, already, in Figure 7b, clear differences in the temporal mass evolution are observed for this particular discharge.

When comparing Figure 8a,b, for any given pressure differential, it is observed that, at time zero, the mass flow, when the fluid goes from L-to-S, is slightly larger than when the fluid goes from the S-to-L, indicating the flow is seeing a higher restriction when the fluid is going from S-to-L reservoir.

Figure 9a,b introduce the Mach number temporal evolution at the nozzle minimum section as a function of the pressure differential and for the two flow directions, L-to-S and S-to-L reservoirs, respectively. The information presented obeys the cases where numerical and experimentally-based results can be compared. When the fluid goes from the large to the small reservoir, the flow is initially sonic, and the time during which the flow remains under sonic conditions increases with the pressure ratio increase, as in Figure 9a. When the fluid is flowing from the small to the large reservoir, and for initial respective pressures of 1.2 MPa and 0.4 MPa, the discharge is sonic during a very small time. But, when the initial reservoirs pressure is of 0.6 MPa and 0.1 MPa, respectively, the time at which the flow remains sonic is of nearly 1 s, which is almost the same time observed when the fluid goes from the large to the small reservoir, and, for a respective pressure of 1.1 MPa and 0.1 MPa, compare figures Figure 9a,b.

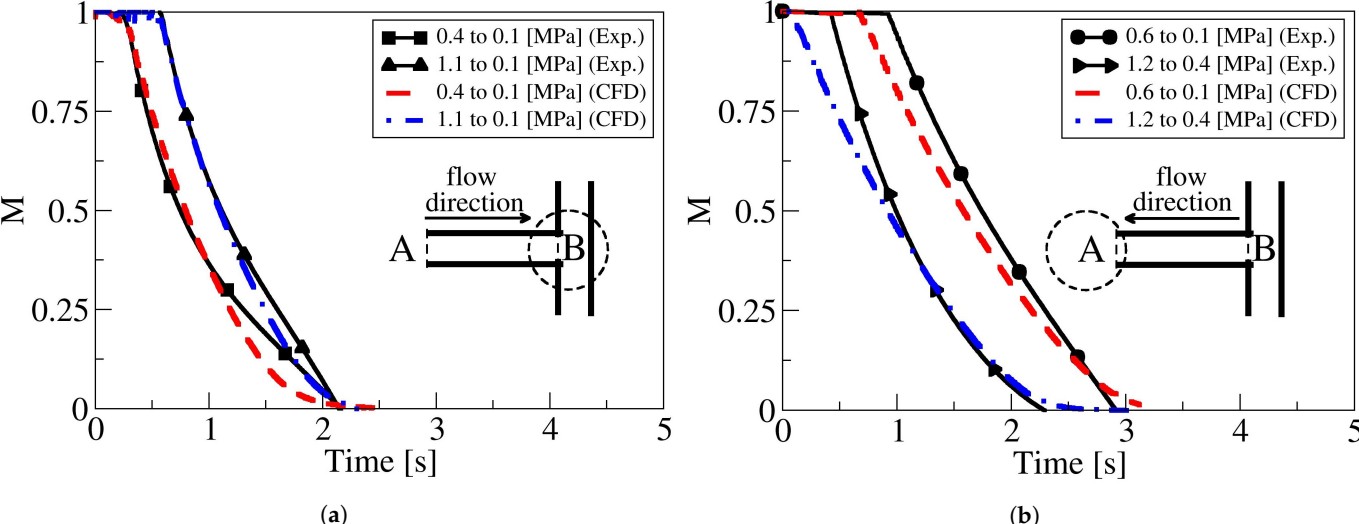

**Figure 9.** Temporal evolution of the maximum Mach number obtained at the nozzle separating the two reservoirs. Comparison between the numerical results and the experimentally-based ones. (**a**) Flow from large to small reservoir. (**b**) Flow from small to large reservoir.

The reason why the fluid remains sonic during a longer time, when the flow goes from S-to-L reservoirs, is likely to be caused by the sudden flow restriction the fluid is suffering when the flow enters the horizontal section of the nozzle and coming from the two T shape branches. The two T shape branches promote the existence of a flow restricted section at the horizontal nozzle inlet, restricting, as well, the entrance of the fluid from a vertical plane, the fluid can only move vertically in the two vertical branches of the T nozzle. In reality, this effect is creating a smaller effective section of the flow in this case than when flow goes from large to small reservoir. In other words, the nozzle resistance to the fluid is larger when the flow goes from the small to the large reservoir; therefore, the mass flow is also smaller. In fact, when comparing the mass flow curves for the same pressure drop presented in Figure 8a,b, it can be clearly seen that the mass flow is higher during the initial times when the fluid goes from the L-to-S reservoir. At this point, it must be highlighted that the location where the Mach number values are computed is always where the spatially averaged Mach number is maximum. Such location is at the end of the horizontal nozzle, beginning of the T junction, when the flow goes from L-to-S reservoir, and at the end of the horizontal nozzle and beginning of the large tank, when the flow goes from the S-to-L reservoir. Such different locations were expected; then, regardless of the

flow direction, the fluid at the entrance of the horizontal nozzle has to be subsonic and accelerates along it.

The temporal discharge coefficients as a function of the Reynolds number, and for the four cases at which CFD and experimentally-based results are generated, is presented in Figure 10. The variations of the discharge coefficient when the flow goes from the large to the small tank, and vice versa, is given in Figure 10a,b, respectively. For both flow directions, the numerical and experimentally-based results are presented for the pressure differentials studied using both methodologies.

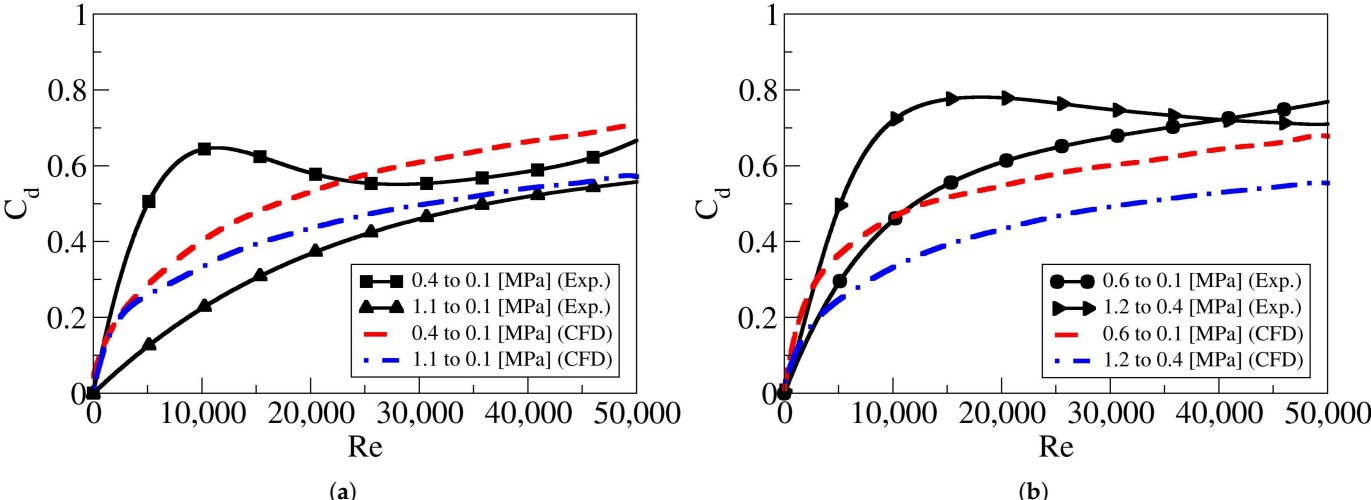

**Figure 10.** Nozzle discharge coefficient based on experimental data and for all pressures studied. (**a**) Flow from large to small reservoir. (**b**) Flow from small to large reservoir.

The first thing to notice is that, for a given flow direction, the discharge coefficient shows a very similar temporal trend, regardless of the pressure ratio evaluated. In fact, it is to be expected that the discharge coefficient depends on the Reynolds number but not on the pressure ratio between reservoirs. Some differences are observed between the discharge coefficients obtained numerically and the ones obtained based on experimental data, particularly at low Reynolds numbers. Authors believe such differences are due to the differences in fluid temperature between experimental and CFD results appearing at the end of the discharge. However, for a given flow direction, the asymptotic values of the discharge coefficients are almost the same regardless of the methodology employed to calculate them. The discharge coefficients obtained when the flow is going from the large to the small reservoir are slightly higher than the ones obtained when the flow is going in the opposite direction. This supports what has been presented until the moment, which is the time required to discharge from the small to the large reservoir is larger than the one needed when the discharge is from the large to the small tank. In other words, the fluid is finding more resistance to flow from the small to the large reservoir than in opposite direction. As explained before, this must be due to the restriction the fluid is observing when flowing from the two T branches and entering the horizontal one.

In order to obtain a single curve representing the evolution of the discharge coefficient as a function of the Reynolds number, and for each flow direction, at each Reynolds number, the average discharge coefficient was determined. The mathematical equation of the resulting curves is presented in Equation (13), which represents the generic equation for the discharge coefficient as a function of the Reynolds number and for both flow directions. The parameters $a_0 \ldots \ldots a_{10}$ characterizing the discharge coefficient curve for each flow direction are defined in Table 2. Notice that, as in Figure 10b, the maximum Reynolds number is 50,000, and the parameters given in the first two rows of Table 2 are valid for this Reynolds number range. Nevertheless, based on the results from the CFD simulations,

a second set of parameters valid for a Reynolds number range $1000 \leq Re \leq 130,000$ are also presented in the last two rows of Table 2.

$$C_d = a_0 + a_1 Re + a_2 Re^2 + \cdots + a_{10} Re^{10}. \tag{13}$$

**Table 2.** Constant values of the discharge coefficient equation for two different range of Reynolds number and different flow directions, large to small reservoirs, and vice versa.

| Reynolds Range | Flow Direction | $a_0$ | $a_1$ | $a_2$ | $a_3$ | $a_4$ | $a_5$ | $a_6$ | $a_7$ | $a_8$ | $a_9$ | $a_{10}$ |
|---|---|---|---|---|---|---|---|---|---|---|---|---|
| 1 to 50,000 | L to S | $1.470 \times 10^{-1}$ | $2.625 \times 10^{-5}$ | $-9.852 \times 10^{-10}$ | $3.077 \times 10^{-14}$ | $-8.131 \times 10^{-19}$ | $1.774 \times 10^{-23}$ | $-2.904 \times 10^{-28}$ | $3.252 \times 10^{-33}$ | $-2.300 \times 10^{-38}$ | $9.184 \times 10^{-44}$ | $-1.572 \times 10^{-49}$ |
| | S to L | $3.608 \times 10^{-2}$ | $1.053 \times 10^{-4}$ | $-5.156 \times 10^{-9}$ | $-1.795 \times 10^{-13}$ | $3.144 \times 10^{-17}$ | $-1.407 \times 10^{-21}$ | $2.351 \times 10^{-26}$ | $1.988 \times 10^{-31}$ | $-1.406 \times 10^{-35}$ | $2.063 \times 10^{-40}$ | $-1.047 \times 10^{-45}$ |
| 1000 to 130,000 | L to S | $1.480 \times 10^{-1}$ | $2.566 \times 10^{-5}$ | $-8.676 \times 10^{-10}$ | $1.981 \times 10^{-14}$ | $-2.541 \times 10^{-19}$ | $7.886 \times 10^{-25}$ | $2.838 \times 10^{-29}$ | $-4.980 \times 10^{-34}$ | $3.824 \times 10^{-39}$ | $-1.481 \times 10^{-44}$ | $2.350 \times 10^{-50}$ |
| | S to L | $6.400 \times 10^{-2}$ | $5.258 \times 10^{-5}$ | $-4.250 \times 10^{-9}$ | $2.382 \times 10^{-13}$ | $-8.453 \times 10^{-18}$ | $1.909 \times 10^{-22}$ | $-2.779 \times 10^{-27}$ | $2.594 \times 10^{-32}$ | $-1.499 \times 10^{-37}$ | $4.884 \times 10^{-43}$ | $-6.857 \times 10^{-49}$ |

One of the advantages of performing 3D-CFD simulations is that it allows to carefully analyze the flow evolution inside the nozzle. The flow field dynamics given as instantaneous velocity contours at both ends of the horizontal pipe, for both flow directions, reservoir pressures and at three different time instants, 0.02 s, 0.5 s, and 1 s, is introduced in Figure 11. Each column characterizes the time at which the velocity field is presented. The initial two rows of Figure 11 show the flow field at both horizontal nozzle ends when the fluid goes from the L-to-S reservoir, and the initial upstream-downstream pressure on each tank is 0.4 MPa–0.1 MPa and 1.1 MPa–0.1 MPa, respectively. The final two rows show the velocity field that, when the flow goes from the S-to-L reservoir, the upstream-downstream initial pressures are 0.6 MPa–0.1 MPa and 1.2 MPa–0.4 MPa, respectively.

When the fluid goes from the large to the small reservoirs, and for the two pressure ratios studied, during the initial milliseconds, $t = 0.02$ s, the fluid reaches sonic conditions at the horizontal nozzle outlet just before the T junction, and supersonic flow conditions are observed as the fluid expands to the two lateral vertical branches; see the first and second rows of Figure 11. At $t = 0.5$ s, and for upstream-downstream initial pressures of 0.4 MPa–0.1 MPa, the fluid has become subsonic at all points, the maximum spatial averaged velocity is 239 m/s, which corresponds to M = 0.74, and the fluid still remains detached when entering the T junction. After 1 s of the origin of the discharge, the maximum fluid velocity has decreased to 120 m/s, but the flow keeps being detached at the T junction entrance. But, when the L-to-S reservoirs initial pressure is of 1.1 PMa and 0.1 MPa, respectively, after 0.5 s, the flow is still under sonic conditions, and the Mach number and the associated spatial averaged velocities at the inlet and outlet of the horizontal pipe are of M = 0.53 (176 m/s) and M = 0.99 (310 m/s), respectively. The respective values of the Mach number in these two pipe locations are of 0.45 and 0.56, after the initial second of the discharge.

The discharge when the fluid flows from the small to the large reservoir at initial pressures of 0.6 MPa and 0.1 MPa, respectively, is presented in the third row of Figure 11. Now, the maximum Mach number appears at the entrance of the large reservoir, and horizontal pipe outlet, as expected according to the theory [29,30]. At time $t = 0.02$ s, the respective Mach numbers and spatial averaged fluid velocities at the horizontal pipe inlet and outlet are M = 0.6 (198 m/s) and M = 1 (353 m/s). For these particular initial pressures, and after 0.5 s, the flow still remains under sonic conditions, and the inlet and outlet Mach numbers and fluid velocities are M = 0.59 (190 m/s) and M = 1 (317 m/s), respectively. At this point, it is important to realize that under sonic conditions, the spatial averaged fluid velocity depends on the instantaneous fluid temperature. When comparing these figures with the ones characterizing the initial pressure drop of 1.2 MPa to 0.4 MPa, presented as the bottom row of Figure 11, it is realized that now just at the initial instants, $t = 0.02$ s, the flow is sonic, but the pressure drop is not large enough to generate a supersonic expansion as the fluid enters the large reservoir. After 0.5 s, the discharge is completely subsonic, being the horizontal pipe inlet and outlet Mach numbers and

associated spatial averaged fluid velocities of M = 0.57 (182 m/s) and M = 0.73 (230 m/s), respectively. At this point, it is interesting to observe the agreement between the CFD results presented in Figures 9 and 11, noticing that the time during which flow is sonic has a perfect match for all pressures studied.

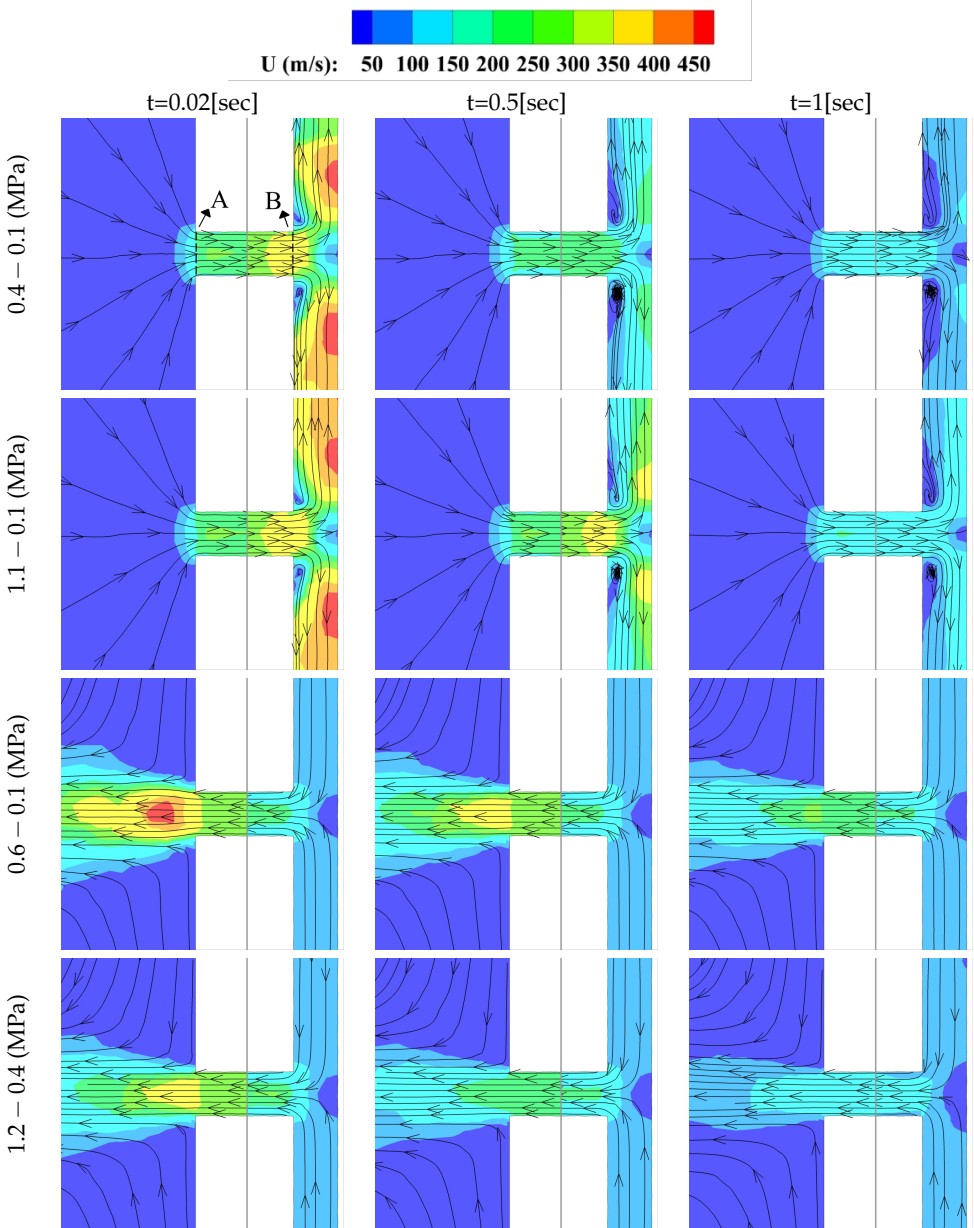

**Figure 11.** Velocity contours of all CFD cases studied and at three instants, 0.02, 0.5, and 1 s. The first and second rows represent the fluid evolution when the fluid goes from the L-to-S reservoir, and the respective upstream/downstream initial pressures are 0.4 MPa–0.1 MPa and 1.1 MPa–0.1 MPa. The third and fourth rows characterize the two cases when the flow goes from S-to-L reservoir, being the upstream/downstream initial pressures of 0.6 MPa–0.1 MPa and 1.2 MPa–0.4 MPa, respectively.

The observations made in Figure 11 are clarifying why, during the experimental tests and CFD simulations, the discharge time was larger when the flow was flowing from the S-to-L reservoir than when flowing in the opposite direction; see all figures between Figures 4 and 8. Notice, as well, from Figure 8, that the maximum mass flow at time zero is always larger when the flow is going from the L-to-S reservoir than when going from the S-to-L one, clearly indicating the added difficulty for the fluid to flow from the small

to the large reservoir. This difficulty can be understood when analyzing the inlet section under both flow conditions. When the flow goes from the large to small reservoir, the flow enters the horizontal nozzle from any direction, 360 degrees, but, when the flow goes from the small to the large reservoir, initially, the fluid needs to enter from the two ends of a T branch and then the fluid needs to enter the horizontal nozzle from the two sides of the T branch, therefore facing a particularly narrow inlet when compared to the opposite fluid direction. The effects of this higher flow restriction, when the fluid is going from the small to the large reservoir, can also be observed when analyzing the discharge coefficients in both flow directions. Notice that the discharge coefficient, when the fluid is going from the S-to-L reservoir, is asymptotically smaller than when the fluid flows from the L-to-S reservoir; see Equation (13) and Table 2.

The work presented in the present manuscript consisted of evaluating the discharge coefficient of a T shape nozzle under compressible flow conditions. Experimental and numerical analyses were performed. Numerical simulations clarified where the sonic conditions are to be expected. Discharge coefficients were dependent on the flow direction and the Reynolds number, and they agree well with the ones obtained by Comas et al. [13] and Nagao et al. [9], specially when considering the different nozzle length to diameter ratio. In the CFD simulations, the fluid was considered as ideal, and similar CFD simulations were performed by Lakzian et al. [18] and Mazzelli et al. [17], where the fluid was considered as ideal, as well, and studied under similar pressure differentials and the same turbulence model. From the comparison of the present study with Reference [9,13,17,18], it can be concluded that the error generated by the CFD simulations is small and acceptable under the engineering applications point of view.

## 6. Conclusions

The discharge time is proved to be directly related to the upstream/downstream density ratio. The discharge coefficients on both flow directions of a T shape nozzle, and considering the fluid as compressible and real, were obtained in the present manuscript based on experimental data. The same information was obtained from CFD simulations. The CFD simulations performed showed a good match with the experimental results and allowed understanding of the differences of the temporal flow evolution inside the nozzle at different flow directions. The exact locations where the flow was sonic, and even supersonic, were detected, allowing further modification of the T shape nozzle design in future applications. The theoretical methodology presented was based on the experimental data and proved to be very accurate and reliable, particularly when temporal pressure and temperature were known. The final equations characterizing the discharge coefficient as a function of the Reynolds number and for both flow directions are provided.

**Author Contributions:** Conceptualization, A.C. and J.M.B.; methodology, C.R.-C., N.M.T., and J.M.B.; experimentation, C.R.-C.; software, A.C. and N.M.T.; validation, N.M.T. and J.M.B.; formal analysis, A.C. and J.M.B.; investigation, C.R.-C., N.M.T., A.C., and J.M.B.; data curation, C.R.-C. and N.M.T.; writing—original draft preparation, N.M.T. and J.M.B.; writing—review and editing, J.M.B.; visualization, N.M.T. and J.M.B.; supervision, J.M.B.; project administration, A.C. and J.M.B.; funding acquisition, C.R.-C., J.M.B., and A.C. All authors have read and agreed to the published version of the manuscript.

**Funding:** Part of the computations were done in the Red Española de Supercomputación (RES), Spanish supercomputer network, under the grant IM-2020-1-0001.

**Institutional Review Board Statement:** Not applicable.

**Informed Consent Statement:** Not applicable.

**Data Availability Statement:** Not applicable.

**Conflicts of Interest:** The authors declare no conflict of interest.

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
