# Peer review of "Discharge Coefficients of a Heavy Suspension Nozzle"

_applsci, doi:10.3390/app11062619_

Round 1

Reviewer 1 Report

An interesting paper using both a numerical and experimental approach to quantify the discharge coefficient of real gas between 2 chambers. The work concluded that the massflow depends first on pressure gradients but also on flow direction.

Detailed feedback is given in the attached pdf in form of comments. Here are the most important comments:

  • please add a few words to define the discharge coefficient in the abstract or the latest in the introduction.
  • Please comment why an ideal gas simulation can be trusted in simulating real gas especially as other author reported a 20% discrepancy between ideal and real gas (line 47)
  • line 157: please add ref for used real gas equations as this depends on the used gas. 
  • figure 3: mesh looks coarse in some areas. A mesh study might backup better your numerical settings.
  • T shape nozzle: in one direction there is mixing of two impinging jets which increases the entropy generation. Figure 11 would have been enough to cover this aspect. The author however showed multiple results as a possible explanation. Please comment on why a T-shape is needed.
  • The results section might be more compact but more important author should guide the reader through the results section and not only introduce figures and tables. please add at least a summary comment at the end of the results section to summarise your results and compare them with the available literature.
  • figure 10 shows a large discrepancy between CFD and experiment. please comment.

Author Response

Please see the file attached.

Reviewer 2 Report

  1. The abstract is not well written. Please rewrite the whole abstract and please make sure to be checked by a native English speaker.
  2. Please specify the CFD software which was used in this study. Was it in-house code or commercial CFD code?
  3. Please add more references to the CFD simulation. What is the current status and what is the main challenge? 
  4. State of the art of CFD simulation should be presented in the Introduction.
  5. Please specify the measurement accuracy for each sensor, e.g., TC, pressure transducer, etc.
  6. The authors should perform the mesh independence study in order to verify the CFD results.
  7. How to generate the mesh? The authors must explain. Did the authors use snappyHexMesh or other mesh generators?
  8. Why did the author choose realizable k-epsilon? Did the authors try another turbulence model? 
  9. How to treat such an unstructured grid in OpenFOAM? It seems that the mesh has a large non-orthogonality value.
  10. What was the time discretization scheme in the present simulation. Please mention this in the manuscript.
  11. How to get the turbulent intensity equal to 0.05%?

Author Response

Please see the file attached.

Reviewer 3 Report

In this paper, the authors evaluated the discharge coefficients of a heavy suspension nozzle using CFD methods and compared them with the experimental data. Good agreements are reached. The overall structure of the paper is good and the conclusions are supported by the results. I have two comments here:

1, The adopted turbulence model is realizable k-e model with wall function. It is surprising to me that it is able to deliver good agreement with the experimental data. I am wondering whether the authors adopted finer mesh and compared the soultion or not. 

2, In figure 6, the predicted scenario of flow goes from small to the large reservoir is bad compared to the other case. Any comments on this?

Author Response

Please see the file attached.

Round 2

Reviewer 2 Report

none